# The Increased Amyloidogenicity of Spike RBD and pH-Dependent Binding to ACE2 May Contribute to the Transmissibility and Pathogenic Properties of SARS-CoV-2 Omicron as Suggested by In Silico Study

**DOI:** 10.3390/ijms232113502

**Published:** 2022-11-04

**Authors:** Anna Y. Aksenova, Ilya V. Likhachev, Sergei Y. Grishin, Oxana V. Galzitskaya

**Affiliations:** 1Laboratory of Amyloid Biology, St. Petersburg State University, 199034 St. Petersburg, Russia; 2Institute of Protein Research, Russian Academy of Sciences, 142290 Pushchino, Russia; 3Institute of Mathematical Problems of Biology RAS, The Branch of Keldysh Institute of Applied Mathematics, Russian Academy of Sciences, 142290 Pushchino, Russia; 4Institute of Environmental and Agricultural Biology (X-BIO), Tyumen State University, 625003 Tyumen, Russia; 5Institute of Theoretical and Experimental Biophysics, Russian Academy of Sciences, 142290 Pushchino, Russia

**Keywords:** SARS-CoV-2, Omicron, Spike, RBD, amyloidogenic properties, molecular modeling

## Abstract

SARS-CoV-2 is a rapidly evolving pathogen that has caused a global pandemic characterized by several consecutive waves. Based on epidemiological and NGS data, many different variants of SARS-CoV-2 were described and characterized since the original variant emerged in Wuhan in 2019. Notably, SARS-CoV-2 variants differ in transmissibility and pathogenicity in the human population, although the molecular basis for this difference is still debatable. A significant role is attributed to amino acid changes in the binding surface of the Spike protein to the ACE2 receptor, which may facilitate virus entry into the cell or contribute to immune evasion. We modeled in silico the interaction between Spike RBDs of Wuhan-Hu-1, Delta, and Omicron BA.1 variants and ACE2 at different pHs (pH 5 and pH 7) and showed that the strength of this interaction was higher for the Omicron BA.1 RBD compared to Wuhan-Hu-1 or Delta RBDs and that the effect was more profound at pH 5. This finding is strikingly related to the increased ability of Omicron variants to spread in the population. We also noted that during its spread in the population, SARS-CoV-2 evolved to a more charged, basic composition. We hypothesize that the more basic surface of the Omicron variant may facilitate its spread in the upper respiratory tract but not in the lower respiratory tract, where pH estimates are different. We calculated the amyloidogenic properties of Spike RBDs in different SARS-CoV-2 variants and found eight amyloidogenic regions in the Spike RBDs for each of the variants predicted by the FoldAmyloid program. Although all eight regions were almost identical in the Wuhan to Gamma variants, two of them were significantly longer in both Omicron variants, making the Omicron RBD more amyloidogenic. We discuss how the increased predicted amyloidogenicity of the Omicron variants RBDs may be important for protein stability, influence its interaction with ACE2 and contribute to immune evasion.

## 1. Introduction

Severe Acute Respiratory Syndrome Coronavirus 2 (SARS-CoV-2) is a highly transmissible, pathogenic Beta coronavirus that triggered a pandemic of acute respiratory disease, also known as coronavirus disease 2019 (COVID-19). Multiple variants of SARS-CoV-2 that differ in their ability to infect human cells and spread in the human population have been described [1,2,3,4,5,6,7]. In addition to the original Wuhan variant, variants of concern, which include Alpha, Beta, Gamma, Delta, and Omicron and are characterized by different transmissivity and/or pathogenicity, have caused consequent SARS-CoV-2 waves in the human population across the world [3]. The SARS-CoV-2 virus as well as its closely related Beta coronavirus SARS-CoV and HCoV-NL63 from the Alpha coronavirus genus use cellular receptor ACE2 (angiotensin-converting enzyme 2) as an entry point [8,9,10,11,12,13]. Virus invasion is dependent on the Spike glycoprotein, which specifically binds to the ACE2 [14]. The Spike forms homotrimers on the viral surface that undergo substantial conformational changes and proteolytic processing (they are cleaved into S1 and S2 subunits) when the virus enters the cell. The S1 subunit contains the receptor binding domain (RBD), which directly binds to the peptidase domain of ACE2, while the S2 is responsible for membrane fusion. Several proteases are responsible for SARS-CoV-2 Spike processing: furin, which cleaves at the S1/S2 site (684–686 a.a.); the transmembrane serine protease (TMPRSS2), which cleaves at S2′ site (815–816 a.a.); and cathepsin L, a pH-dependent endosomal protease, which mediates the endosomal entry route [8,15,16,17]. The three RBDs are at the apex of the Spike trimer and can be positioned in the “up” conformation accessible for receptor binding and the “down” conformation, occluding the Spike from binding. While entering the cell, the Spike protein undergoes remarkable structural changes that lead to S1 subunits disengagement and S2 trimer elongation [18].

SARS-CoV-2 can cause respiratory system damage, endothelium dysfunction, thrombo-inflammation and multiple organ failure [19,20,21]. A growing body of evidence suggests a strong link between COVID-19 and central nervous system (CNS) disorders. Several studies have shown that SARS-CoV-2 infection increases the risk for neurodegenerative diseases [22,23,24,25,26,27]. Structural changes in the brain associated with COVID-19 have been confirmed in both surviving patients and non-survivors and may be caused by different mechanisms [25]. Amyloid deposition caused by SARS-CoV-2 may play a role in virus pathogenicity and affect different organs and tissues [25,28,29,30]. SARS-CoV-2 infects mature neurons and exacerbates Aβ aggregation and Alzheimer’s neuropathology [26,31]. This is of particular interest since Aβ peptides have demonstrated antimicrobial and antiviral activity, e.g., against influenza A virus and Herpes simplex virus 1 (HSV-1) [32,33,34]. It is hypothesized that Aβ deposition in the brain may be triggered by pathogenic infections including SARS-CoV-2. Some viruses such as HSV-1 can directly catalyze the aggregation of Aβ42 [35]. Interestingly, some researchers attribute this effect to the heparin-binding abilities of these viruses and include SARS-CoV-2 in this group [24,36]. Heparins are long, unbranched, and highly negatively charged glycosaminoglycan molecules that are closely associated with different amyloid fibrils isolated from humans and are thought to play a role in amyloid fibril formation and stabilization [37]. Surface plasmon resonance and circular dichroism experiments have shown that the S1 and specifically the RBD-domain (residues 330–583) interacts with heparin, and this interaction causes conformational changes in the RBD-domain [38]. This is supported by docking experiments showing S1 RBD interaction with heparin and heparin-binding proteins (HBPs) [39,40]. For instance, S1 RBD was speculated to bind to several heparin-binding amyloidogenic proteins including Aβ peptides, α-synuclein, tau, TDP-43 RRM, and PrP prion protein from turtle [39]. The heparin-binding site on the Spike surface might serve as a docking site for binding to the Aβ42 peptide and promoting catalyzation of Aβ42 aggregation via surface-assisted heterogeneous seeding [24]. Aβ42 binds with high affinity to both the S1 of SARS-CoV-2 and ACE2, which may play an important role in the development of severe COVID-19 [41]. The interactions between SARS-CoV-2 Spike and ACE2 have also been suggested to promote the spread of cytosolic prions and Tau aggregates [42]. Thus, the Spike protein is discussed as a cross-seeding platform that promotes the aggregation of heterologous amyloid-like proteins, which can lead to the production of toxic amyloid aggregates in the intracellular or extracellular space.

Equally interesting is the potential of the Spike for self-assembly. Various fragments of the Spike protein have been suggested to form amyloid-like structures. Amyloidogenic regions have been detected in the Spike protein using several algorithms including AGGRESCAN, Waltz, and FoldAmyloid [24,33,43]. The short peptide RSAIEDLLFDKV, which immediately follows the second cleavage site in the S2 domain and is relatively conserved in many coronaviruses, including SARS and MERS, forms amyloid-like β-sheet fibril structures [44]. Finally, Spike amyloid-like aggregation has been shown upon lipopolysaccharide binding [45]. Spike protein can be processed by several proteases in addition to furin-like proteases and TMPRSS2, which can expose multiple amyloidogenic segments in the proteolytically nicked Spike protein. Interestingly, S-protein proteolysis by Neutrophil Elastase renders amyloid-like fibrils [43]. In addition, a recent in silico study has detected prion-like domains in the SARS-CoV-2 Spike protein RBD [46]. This observation is particularly relevant to the fact that SARS-CoV-2 exhibits a 10-fold higher affinity to the ACE2 receptor than SARS-CoV [47]. The authors have found a significant difference in the prion-like properties of the Spike protein in novel emerging SARS-CoV-2 variants, with an increase in the prionogenecity for Delta (B.1.617.2) and a decrease for Omicron (B.1.1.529) [46]. Mohabatkar et al. have presented in silico methods for predicting the prion-like domain in the Nsp3 protein of SARS-CoV-2 [48]. However, the exact role of prion-like domains associated with viral evolution is still unclear [49].

ACE2-binding affinity of the Spike is believed to be one of the most important determinants of SARS-CoV-2 infectivity and disease severity [50]. Other important factors are the utilization of cellular proteases for Spike processing and syncytia formation. Some studies have found that Omicron demonstrated higher ACE2 binding and outcompeted Delta when replicating in human nasal epithelial cells [51,52]. At the same time, its replication in lung epithelial cells is reduced compared to Wuhan D614G and Delta [52,53,54]. Notably, Omicron demonstrated suboptimal S1/S2 cleavage compared to Delta, and its entry into the cell seems to be independent of TMPRSS2 [52,53]. In another recent report, Omicron Spike was found to have compromised fusion activity due to structural changes, so higher levels of ACE2 are required for efficient cell entry [55]. While the affinity to ACE2 is expected to be modulated by mutations affecting Spike residues involved in the direct recognition of ACE2, the same could be achieved by affecting structure-forming properties of the protein. Cryo-EM structural analysis of the Omicron Spike trimer at pH 5.5 and pH 7.5 showed that acidic pH provoked structural changes within Receptor Binding Motif (RBM) [56]. No less interesting is that electrostatic changes occurred in the Omicron RBD when compared to the prototype Wuhan-Hu-1 [57].

Molecular Dynamics (MD) simulation is a powerful tool allowing the detailed characterization of structural features from a dynamic perspective and has effectively been used for SARS-CoV-2 studies [58]. Here, we used a computational approach to analyze the amyloid-like properties of the Spike RBDs of several SARS-CoV-2 variants (original Wuhan-Hu-1 variant (WT), Alpha, Beta, Delta, Epsilon, Gamma, and Omicron BA.1 and BA.2 variants) and to investigate a link between RBD amyloidogenicity and strength of interactions with ACE2. We noted that the SARS-CoV-2 Spike RBD evolved into a more charged, basic surface with two longer amyloidogenic tracts. MD simulations at various pHs showed that the strength of the interaction between RBD and ACE2 was higher for Omicron BA.1 RBD compared to Delta RBD or Wuhan-Hu-1 RBD. The strongest binding was observed for protonated Omicron BA.1 form. We hypothesize that the more basic surface of the Omicron variant may be beneficial for its spread in the upper respiratory tract, but not in the lower respiratory tract, where pH estimates are different. We also propose that the increased amyloidogenicity of Omicron variants may confer greater stability, influence its interaction with ACE2, and affect immune evasion.

## 2. Results

### 2.1. Omicron RBD Better Binds to ACE2 in Protonated form When Compared to Delta by MD Simulations

During its spread in the human population, SARS-CoV-2 has gained many mutations that influence its binding to the ACE2 receptor, affect proteolysis, or impact its immune evasion. For instance, Omicron has gained mutations that affect the binding interface of Spike with ACE2 (Figure 1a), although their impact on the binding affinity is still debatable [55,56,57,58,59,60,61,62,63,64,65,66]. The detailed structural characteristics of the Omicron Spike protein’s interaction with ACE2 showed that substitutions T478K, Q493R, G496S, and Q498R may establish hydrogen bonds or salt bridges with ACE2 [56,57]. Along with the N501Y mutation, which is a known determinant of increased binding affinity, these mutations can enhance Omicron binding affinity to ACE2 [56,66]. Importantly, T478K, Q493R, and Q498R substitutions significantly increased the positive charge while E484A decreased the negative charge of Omicron RBD [57] (Figure 1a).

We compared the RBD and the RBM sequences of Wuhan-Hu-1, Alpha, Beta, Delta, Epsilon, Gamma, Omicron BA.1, and Omicron BA.2 variants, analyzed their charge at pH 7 and the isoelectric point (Table 1). We found that charge and pI consistently increased from Wuhan-Hu-1 to Omicron variants, and this increase correlated with virus transmissivity. Evidently, the SARS-CoV-2 RBD has evolved towards a more charged, basic surface. The increase in the basic amino acid residues (K and R) is seen mainly within the RBM, indicating the importance of these changes for interaction with ACE2. We hypothesized that these changes might influence the interaction between the Omicron RBD and ACE2 at different pH levels.

To analyze in silico the pH impact on the binding affinity of Spike RBDs to ACE2, we modeled the interaction of RBDs of variants Wuhan-Hu-1, Delta, and Omicron BA.1 with the ACE2 receptor at pH 5 and pH 7 at speeds 0.1, 0.05, and 0.01 Å/ps (Appendix A). We noted no difference between variants at the speed 0.01 Å/ps; however, at speeds 0.1 and 0.05 Å/ps, RBD variants showed different dynamics. As seen in Figure 1b, the affinity of Omicron RBD to the ACE2 is higher than that of Wuhan-Hu-1 and Delta RBDs at both pHs, pH 5 and pH 7. The difference between protonated and unprotonated forms is statistically significant at both speeds, 0.1 and 0.05 Å/ps, for Omicron BA.1 only (Figure 1b). Omicron BA.1 RBD in its protonated form showed the strongest interaction with ACE2 among all variants we evaluated (Figure 1b). We did not find any statistically significant difference between Wuhan-Hu-1 RBD and Delta RBD at any pH or modeled speed.

Appendix A and Appendix A show the values of the potential energy of the system for the interaction between RBDs and ACE2 during the whole period of relaxation and at the end of 9 ns relaxation. The lower the potential energy value, the greater the force needed to tear one domain from another. We observed that the full potential energy of the system is the lowest in the case of protonated Omicron BA.1 RBD—ACE2, thus confirming our conclusion that Omicron BA.1 in the protonated form has the strongest interaction with ACE2 (Appendix A). Figure 1c shows the force reactions for the SARS-CoV-2 RBD variants and ACE2 during one realization of experiments for unprotonated variants. The force reaction is the highest for Omicron BA.1 RBD compared to Wuhan-Hu-1 and Delta RBDs. The same trend was observed for 0.05 Å/ps speed (Appendix A).

Therefore, our calculations suggest that (i) Omicron BA.1 RBD binds better to ACE2 compared to Wuhan-Hu-1 RBD and Delta RBD; (ii) all variants’ RBDs tend to bind better to ACE2 in the protonated form (at lower pH); (iii) Omicron RBD showed the strongest binding to ACE2 in the protonated form, which was statistically significant compared to unprotonated at two speeds, 0.1 and 0.05 Å/ps, among all variants we analyzed.

It is noteworthy that pH differs in the upper respiratory tract and in the lungs: in the upper respiratory tract, it is more acidic and ranges from 5.5 to 6.5, while in the lungs, it is higher, closer to 7.6 [73,74]. Thus, a more charged Omicron RBD surface may more easily bind to ACE2 in the upper respiratory at a pH below 6.5, but is less favored at higher pH. Thus, our data can explain the efficient transmission of Omicron in the population combined with its apparent lower pathogenicity. Omicron can easily spread in the upper respiratory tract, but damages the lungs and endothelium less because the pH is less than optimal for its binding to ACE2 there.

### 2.2. Amyloidogenic Properties of Spike RBDs of SARS-CoV-2 Variants

Amyloidogenicity is considered an important factor that might affect host–pathogen interaction and impact the virulence and pathogenicity of bacteria and viruses [35,75]. We compared the RBD sequences of Wuhan-Hu-1, Alpha, Beta, Delta, Epsilon, Gamma, Omicron BA.1, and Omicron BA.2 variants and computed their amyloidogenic potential (Table 2, Figure 2a). We found that Delta and Epsilon RBDs have a reduced amyloidogenic potential (29.7%) compared to the Wuhan-Hu-1 variant (34.4%). At the same time, amyloidogenicity was increased for Omicron BA.1 (38.3%) and Omicron BA.2 (40.7%) (Table 2, Figure 2a). Surprisingly, the main changes occurred in regions 366–381 a.a. and 487–497 a.a., where longer amyloidogenic tracts were predicted by FoldAmyloid for Omicron BA.1 and Omicron BA.2 but not for other variants (Figure 2b).

The amyloidogenic region 487–497 a.a. lays within the RBM represented by a variable region 438–506 a.a. and overlaps with the T470-F490 loop, which, along with Q498-Y505 region, is a critical element in ACE2 recognition [77,78,79]. The remaining portion of the RBD functions as a structural core stabilized by the antiparallel β-sheet [3,47,78,79,80]. The two mutations, Q493R and G496S, that affect the ACE2 binding are in the longer Omicron BA.1 amyloidogenic tract and Q498R is just adjacent to it (Figure 2b)

**Figure 2 ijms-23-13502-f002:**
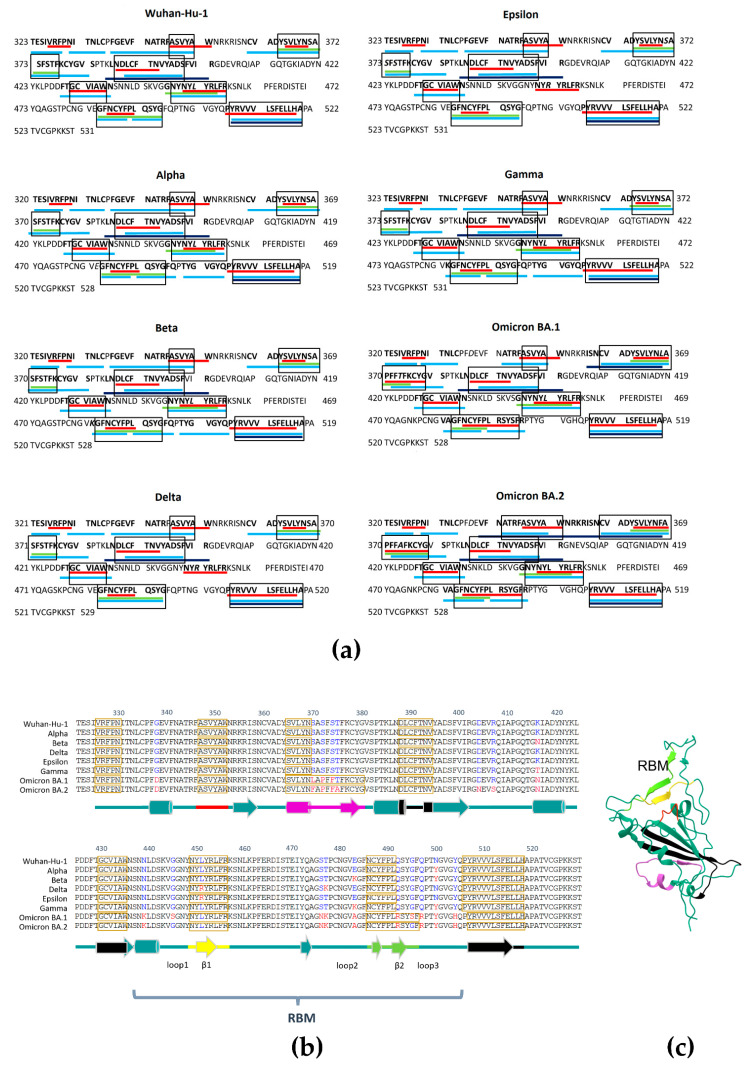
(**a**) Amyloidogenicity of Spike RBDs of different variants of SARS-CoV-2 as assessed by FoldAmyloid (red), Waltz (green), AGGRESCAN (light blue), and PASTA 2.0 (dark blue). (**b**) Alignment of SARS-CoV-2 variants’ Spike RBDs with amyloidogenic regions (boxed) predicted by FoldAmyloid. The secondary structure of Spike RBD is shown in cyan under the sequence. Amyloidogenic tracts 348–353 a.a. (red), 366–381 a.a. (magenta), 388–395 a.a. (black), 431–436 a.a. (black), 450–457 a.a. (yellow), 487–497 a.a. (light green), and 508–519 a.a. (black) are highlighted. Loop 1, Loop 2, Loop 3, β1, and β2 are as in [81]. (**c**) The 3D architecture of Omicron RBD according to the structure 7T9L (https://www.rcsb.org/3d-view/7T9L/1, accessed on 20 June 2022) with amyloidogenic tracts colored as in (**b**).

Figure 3a shows the position of these changes on the 3D Omicron RBD structure with amyloidogenic tracts highlighted in colors. Amyloidogenic region 366–381 overlaps with the lipid-binding pocket, which was seen in a locked-down configuration upon linoleic acid or similar fatty acid binding and may play a role in Spike aggregation stimulated by lipopolysaccharides [45,67,68]. Another interesting point is the aromatic interaction between neighboring RBDs subunits involving Y369/F374 from the RBD structural core and F486/Y489 of RBM [77]. This interaction was proposed to enhance the interaction between S1 subunits [77]. We noted that Y369 and F374 residues are within the longer amyloidogenic tract in the Omicron BA.1 and Omicron BA.2, where Y and F residues within this tract are conserved in the Omicron variants (Figure 2b). The position of these amino acids relative to the 3D Omicron BA.1 RBD structure is shown in Figure 3b. We also noticed that amyloidogenic tract 348–353 a.a. is located in close proximity to the amyloidogenic tract 450–457 a.a. (Figure 2b,c, red and yellow tracts correspondingly), which may determine their interaction and stabilization of RBM in a conformation that favors the interaction with ACE2 (see also Figure 3a,b).

## 3. Discussion

The transmissibility of the SARS-CoV-2 variants has dramatically increased during its evolution in the human population. Earlier estimates of the basic reproduction number (R0) of SARS-CoV-2 in China were between 2.2 and 3.6 [82,83]. For the Delta variant, this number ranged from 3.2 to 8 and more than three times in the Omicron variants [71,84]. The transmissibility of the Omicron BA.2 variant was estimated to be approximately 1.4 times higher than that of BA.1 [72]. SARS-CoV-2 has accumulated many mutations during its spread, and many of them are located in the RBD of the Spike protein, affecting some of the key positions interacting with amino acid residues of the ACE2 receptor, which was generally believed to influence its binding properties to the ACE2 receptor. For instance, nine of them, K417N, G446S, S477N, E484A, Q493R, G496S, Q498R, N501Y, and Y505H, are located at the ACE2 binding interface, while T478K is located just at the periphery (Figure 1a) [57,64,85]. Despite this fact, reports comparing the binding affinity of Delta and Omicron RBDs to the ACE2 are still controversial. While some researchers observed binding affinity of Omicron RBD comparable to that of Wuhan (WT) RBD or weaker than that of Delta RBD [57,61,64,65], others report increased binding of Omicron RBD to ACE2 compared to WT RBD [56,63]. The binding affinity of Omicron to ACE2 may depend on various factors such as the type of infected cells and conditions for viral replication. Thus, Omicron demonstrated higher ACE2 binding and outcompeted Delta when replicating in human nasal epithelial cells but not in lung epithelial cells [51,52,53,54]. In addition, analysis of binding free energy (BFE) changes in the structure of the RBD−ACE2 complex, reflecting viral infectivity, showed that the Delta variant had the highest BFE change among the earlier variants, and the Omicron variants had BFE changes that were about 1.5–2 times higher than those of Delta, implicating Omicron variants and specifically Omicron BA.2 as being the most contagious [86]. In our MD simulations of RBD binding to ACE2, we observed that Omicron RBD binds better than Wuhan-Hu-1 RBD and Delta RBD, and this effect was the most significant for the protonated form. Notably, structural changes in the Omicron Spike RBM have been noted at acidic pH in Cryo-EM analysis [56]. Combined with our data, this reinforces the idea that Omicron RBD can exhibit different binding properties at the acidic, neutral, or slightly alkaline pH values. Notably, the pH dependence of virus transmissibility has been described for the H5N1 influenza virus [87,88]. The decreased hemagglutinin protein activation pH enhanced the growth of the H5N1 influenza virus in the mammalian upper respiratory tract [87].

Amyloidogenic regions in Spike protein have been evaluated earlier using different algorithms [24,33,43,44,45]. For instance, seven peptides covering regions 191–210 a.a., 259–279 a.a., 362–381 a.a., 531–551 a.a., 599–618 a.a., 688–708 a.a., and 1165–1184 a.a. were predicted to be amyloidogenic using Waltz [43]. Three of these peptides spanning 191–210 a.a., 599–618 a.a., and 1165–1184 a.a. met the general amyloid criteria; i.e., they demonstrated sigmoidal polymerization kinetics with Thioflavin T, polarization upon Congo Red staining and fibrillar ultrastructure [43]. Some of the amyloidogenic peptides might be released after proteolytic cleavage at S1/S2 and S2′ sites [24,44]. Here, using a computational approach, we evaluated amyloidogenic regions in RBDs of various SARS-CoV-2 isolates and found that Omicron variants have two longer amyloidogenic tracts encompassing the regions 366–381 a.a. and 487–497 a.a.

The cryo-EM and crystal structures of RBD binding to ACE2 specified that T470-F490 and Q498-Y505 regions at the end of the S1 RBD are key contact elements with ACE2 with T470-F490 loop being flexible between open and closed states [77,78,79]. In addition, the MD computational study has proposed the “anchor-locker” recognition mechanism of RBM binding to ACE2 [81]. In this model, Loop 2 (Anchor, coincides with critical T470-F490 element) and Loop 3 (Locker, partially overlaps with Q498-Y505) are significantly involved in ACE2 binding and β-strand 1 region reinforced binding after the recognition at Loops 2 and 3 on both sides. Two β-strands, β-strand 1, and β-strand 2 are usually in native contact at the binding interface. Interestingly, the region 487–497 a.a. with increased amyloidogenicity overlap with the Loop 2-β-strand 2-Loop 3 structure (Figure 2b,c). Therefore, it is of great interest whether the increased amyloidogenicity of the binding interface may contribute to the kinetics of RBM binding to the receptor and/or play a role in Omicron’s immune evasion. The Q493R mutation, which lies within the 487–497 a.a. region with increased amyloidogenicity (Figure 2b and Figure 3a) is considered one of the key immune escape sites in Omicron, which are critical for neutralizing the activity of the monoclonal antibodies bamlanivimab and etesivimab [63,64,89,90,91,92]. No less interesting is S371L mutation located in the Omicron BA.1 amyloidogenic tract 366–381 a.a. (Figure 2b and Figure 3a), which affects the recognition of a wide range of monoclonal antibodies [64,92,93]. Approximately 10% of variants of B.1.1.529 (Omicron) lineage contain additional mutation R346K adjacent to the 348–353 amyloidogenic tract. Interestingly, R346K renders almost all current antibody therapy for COVID-19 ineffective, highlighting the importance of this epitope in virus adaptation and evolution [92,93].

The 366–381 region with increased amyloidogenicity overlaps with the lipid-binding pocket of Spike RBD. It was speculated that S371L, S373P, and S375F mutations may affect the flexibility of the lipid-binding pocket and change the lipid-binding properties of Omicron [64]. On the other hand, amyloids are known to interact with lipids and lipid bilayers, and this binding may promote amyloid fibril nucleation [94,95]. The 366–381 amyloidogenic regions overlap with Spike 365 peptide KKKGGGYSVLYNSASFSTFK, forming amorphous aggregates stained by Congo Red [43]. In addition, Petrlova et al., in a recent study, showed Spike amyloid-like aggregation upon binding to lipopolysaccharide, which may implicate the same region [45].

Another interesting observation is that the amyloidogenic tracts affect regions that are thought to be involved in S trimer conformational transitions from the ground prefusion state toward the postfusion state. In the prefusion state, the RBDs are in triple—“down”–conformation, which upon binding to the ACE2, has been suggested to adopt single—“up"—conformation [77,96]. Double—and triple—“up” conformations were observed more rarely in the structural studies. For SARS-CoV, they represent 39% and 3%, respectively [97]. Xu et al. proposed the interaction between the core RBD region and the RBM 470–489 loop of neighboring S1 subunits in single—“up”—conformation involving Y369/F374 and F486/Y489 aromatic interaction [77]. Based on these observations, it is expected that the two longer amyloidogenic tracts 366–381 a.a. and 487–497 a.a. in Omicron variants may be in direct contact with each other. It is of interest whether the interaction between them might help to stabilize the S protein in the “up” conformation in the trimeric structure of the Spike protein. It was observed that the Omicron Spike trimer exhibits a much more compact architecture than Delta or WT Spike trimers and improved contacts between S2-S2 and S1-S1 subunits. The thermal stability of the Omicron Spike trimer was increased compared to WT and Delta [56].

Interaction with other proteins can be influenced by the Spike amyloidogenic tracts. It has been shown that Spike induces structural changes in β and γ fibrin(ogen), complement 3, and prothrombin, increasing their resistance to trypsinization, which, among other mechanisms, may contribute to hypercoagulation in patients with COVID-19 [98]. A recent preprint suggests that SARS-CoV-2 induces amyloid aggregation of several proteins involved in neurodegenerative diseases such as APLP1, ApoE, clusterin, α2-macroglobulin, PGK-1, ceruloplasmin, nucleolin, 14–3-3, transthyretin, and vitronectin [99]. Patients hospitalized with COVID-19 receive anticoagulant therapy to prevent thrombosis, including low-molecular-weight heparin [100,101,102]. Besides its blood-thinning ability, heparin is known for its association with various amyloid fibrils. Interestingly, surface plasmon resonance and circular dichroism spectroscopy demonstrated that heparin can directly bind to and induce conformational changes in the Spike RBD of SARS-CoV-2. In cell culture assay, it prevented SARS-CoV-2 invasion of Vero cells by up to 80% [38].

A recent in silico study has detected the prionogenic domain in the SARS-CoV-2 RBD that overlaps with the loop2-β2-loop3 structure. Among the various virus variants, the Delta Spike protein showed the highest prionogenic scores, while Omicron Spike had the lowest one [46]. The PLAAC prion prediction algorithm used by the authors is based on the amino acid frequencies in the subset of Prion Domains of the yeast *Saccharomyces cerevisiae* [103]. Yeast prionogenic tracts are usually enriched with polar uncharged Q and N residues, while negatively and positively charged amino acids (D, E, R, and K) are underrepresented [104]. Q- and N-rich prion domains per se can be classified as a distinct type of amyloidogenic sequences and are not scored by FoldAmyloid due to the packing density of Q and N below the threshold [105,106,107] using the scale of the expected number of contacts but can be predicted using the scale of hydrogen bonding. Therefore, the increased amyloidogenicity is expected to be inversely correlated with the PLAAC-predicted prionogenicity. We speculate that the increased prionogenicity of Spike may be associated with the pathogenic properties of SARS-CoV-2 (e.g., the Delta variant is associated with an increased risk of hospitalization and higher mortality), while the increased amyloidogenicity may be associated with the transmissibility of Omicron BA.1 and BA.2 variants and lower pathogenicity. It is of interest that the seasonal respiratory Alpha coronavirus HCoV-NL63, also entering the cell through the ACE2 receptor route, demonstrated a very high degree of amyloidogenicity in its RBD (Table 2), further supporting the hypothesis that RBD amyloidogenicity may be associated with elevated stability of Spike proteins in open conformation and transmissibility of Alpha and Beta coronaviruses.

## 4. Materials and Methods

The Spike protein is a homotrimer where each SARS-CoV-2 Spike protein subunit consists of 1273 amino acids. The sequences of the SARS-CoV-2 Spike RBDs were of Wuhan-Hu-1, YP_009724390.1 (residues 323–531), Alpha B.1.1.7, QWE88920.1 (residues 320–528), Beta B.1.351, QRN78347.1 (residues 320–528), Delta B.1.617.2, QWK65230.1 (residues 321–529), Gamma P.1, B.1.1.28.1, QVE55289.1 (residues 323–531), Epsilon B.1.429, QQM19141.1 (residues 323–531), and Omicron BA.1 (previously B.1.1.529), UFO69279.1 (residues 320–528), and Omicron BA.2, UJE45220.1 (residues 320–528). Wuhan-Hu-1 Spike amino acid sequence was used as a reference for amino acid position indication.

The pI and charge at pH 7 were calculated using the program “Protein Calculator v3.4” available online: http://protcalc.sourceforge.net/ (accessed on 24 June 2022).

For molecular modeling, we used PDB structures 6m0j (Wuhan-Hu-1), 7w9i (Delta), and 7t9l (Omicron BA.1). Each PDB structure contains the RBD domain of the Spike protein and the ACE2 receptor. There is no valence bonding between RBDs and ACE2. The software packages PUMA [108,109] and PUMA-CUDA were used as an MD Modeling Program. PUMA-CUDA has great performance due to the use of various parallel programming technologies (parallel operation on multiprocessor systems with shared memory, with distributed memory, as well as work on graphics accelerators, GPGPU). The AMBER [110] force field and the TIP3P water model were used [111]. The resulting trajectories of molecular dynamics were investigated by the Trajectory Analyzer of Molecular Dynamics TAMD [112,113,114].

The initial coordinates of all systems were taken from the Protein Data Bank (https://www.rcsb.org/, accessed on 1 February 2022). The 15 Å layer of water molecules was added near protein structures. A collisional thermostat was used during all MD experiments for temperature maintenance near 300 K. The main integrity step was 0.001 ps^−1^.

First, all three structures were relaxed for 1 ns to eliminate coordinate inaccuracies and stressed areas. The basis of the experiments is force unfolding. With force separation, the strongest sample will show the greatest force reaction. There are two material points in the system: one in the center of mass of the first system (ACE2) and the second in the center of the RBD. The first system is fixed. The second system breaks away from the first due to the movement of the second material point at a constant speed along the axis, connecting the center of mass of the systems. The reaction force is measured as the force of the Hooke spring connecting the second material point to the second system. The speed is chosen as 0.1, 0.05, and 0.01 Å/ps. With each speed of force separation of structures, eight independent computational experiments were carried out from the same initial data. The main characteristics of 144 MD simulations are available at the site: http://oka.protres.ru/protres_sars, accessed on 19 October 2022. A video of pulling the RBD domain of the Wuhan strain from ACE2 is available at the http://oka.protres.ru/protres_sars/Video/Expand-6m0j.avi, accessed on 19 October 2022, and that of the Omicron domain is available at the http://oka.protres.ru/protres_sars/Video/Expand-7t9l.avi, accessed on 19 October 2022.

The 3D images of Omicron Spike RBD in complex with ACE2 were created using PDB 7t9l (https://www.rcsb.org/structure/7T9L, accessed on 20 June 2022) and Mol*Viewer [115].

The amyloidogenic properties of the proteins were estimated using the FoldAmyloid program [105]. Additionally, we used Waltz, Aggrescan, and PASTA 2.0 [116,117,118]. For the theoretical identification of consensus amyloidogenic regions, the standard program settings recommended by the developers were used. Consensus sequences were chosen based on the coincidence of the prediction results of at least two programs. Protein amyloidogenicity was calculated as a percentage of the number of amino acid residues included in consensus amyloidogenic regions to the total number of amino acid residues of the protein.

## 5. Conclusions

Studying the structure and behavior of the SARS-CoV-2 Spike protein is an important step in the development of effective therapeutics against SARS-CoV-2. Despite extensive analysis involving elaborated structural and computational methods, many aspects of Spike’s interaction with ACE2 remain controversial or poorly understood. SARS-CoV-2 is rapidly evolving in the human population, with new variants constantly emerging, some of which are superior to their predecessors. The mechanisms that provide the selective advantage of new SARS-CoV-2 variants are not fully understood. SARS-CoV-2 can accumulate mutations that improve its binding to the receptor, facilitate Spike processing, or structural changes important for its entry into the cell. Many mutations have accumulated in the RBD, most often in the RBM of Spike. We noted that over the course of its evolution, the virus has gained a more charged RBD, and its receptor binding surface has become more basic compared to early variants of SARS-CoV-2. Using molecular modeling, we found that in its protonated form, Omicron BA.1 showed the strongest binding to ACE2 compared to Delta or Wuhan-Hu-1. We hypothesize that this may explain the high transmissibility and rapid spread of Omicron BA.1 in the population, as this variant seems to be adapted to the slightly acidic environment of the upper respiratory tract compared to its predecessors. Our study has limitations based on its in silico nature, and direct experiments measuring the binding of Spike RBDs of various SARS-CoV-2 variants to ACE2 at different conditions including different pHs may help to better elucidate the molecular basis of Omicron spread in the population.

Using the FoldAmyloid program, we found eight amyloidogenic regions in the RBDs of all analyzed Spike variants. Notably, two of these regions were longer in the Omicron BA.1 and BA.2 variants. These regions influence important structural elements of the RBD: the region involved in ACE2 binding (overlapping with the amyloidogenic tract 487–497 a.a.) and the region important for lipid binding (overlap with amyloidogenic tract 366–381 a.a.). These two amyloidogenic tracts may be in contact with each other, enhancing the interaction between different RBD protomers within the Spike trimer, affecting the lipid binding and the conformational state of Spike. Therefore, we suggest that the amyloidogenic properties of RBD Omicron Spike may be important determinants influencing its conformation and receptor binding. Importantly, two amino acid substitutions, Q493R and S371L, critical for Omicron’s immune system evasion, are found in these two long amyloidogenic tracts. More studies are needed to establish whether the increased amyloidogenicity of Omicron Spike RBD contributes to SARS-CoV-2 spread in the population and immune evasion.

## Figures and Tables

**Figure 1 ijms-23-13502-f001:**
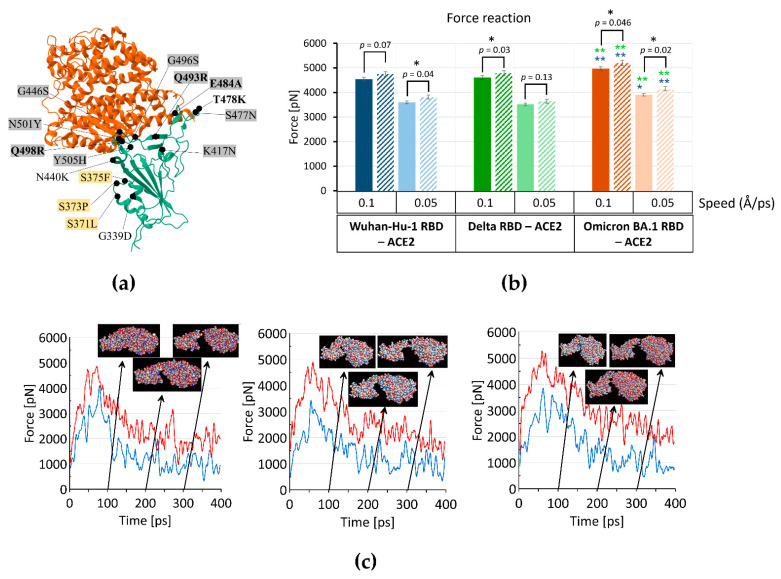
(**a**) Cryo-EM structure of SARS-CoV-2 Spike Omicron BA.1 RBD (cyan) with ACE2 domain (orange) (https://www.rcsb.org/3d-view/7T9L/1, assessed on 1 June 2022). Positions mutated in Omicron BA.1 are shown as black circles. Grey boxes indicate mutations that are at the binding interface with ACE2, N440K, and T478K are just peripheral and might influence the binding, and yellow boxes indicate mutations that affect the lipid-binding pocket of Spike RBD [45,64,67,68]. Bold—mutations within RBM at the binding interphase with ACE2 or adjacent to it that increase the positive charge (T478K, Q493R, Q498R) or decrease the negative charge (E484A) according to [56,57]. (**b**) The force reaction in steered Molecular Dynamics simulations of separating the RBD domain from the ACE2 with constant speed 0.1 or 0.05 Å/ps (indicated below the x-bar) of different SARS-CoV-2 variants: Wuhan-Hu-1 (6m0j, dark and light blue), Omicron BA.1 (7t9l, dark and light red), Delta (7w9i, dark and light green), and each protonated form (diagonal line pattern). The mean of eight independent experiments is shown in each case, and the standard error of the mean is shown as error bars. The *t*-test was used for comparing the means, and *p*-levels are indicated above the bars; *—denotes a significant difference at the *p*-level < 0.05, **—denotes significant difference at the *p*-level < 0.01. Blue and green asterisks above the Omicron BA.1 bars correspond to significant differences between the corresponding bars of Omicron BA.1 RBD and Wuhan-Hu-1 RBD (blue) and Omicron BA.1 RBD and Delta RBD (green). (**c**) Timeline (Force pN/Time ps) of pulling the RBD of Wuhan-Hu-1, Omicron BA.1, and Delta with constant speed 0.1 Å/ps from the ACE2 domains, all unprotonated variants. Force reaction of ACE2 (blue) and Spike RBD (red) domains. Three-dimensional-structures at the top of the figure correspond to 100, 200, and 300 ps of trajectory. At the end of the pulling, last destructed contacts were for 6m0j: S19 and T27 from ACE2 and S477 and F486 from RBD; for 7t9l: E23 and T20 from ACE2 and N477 and F486 from RBD; for 7w9i: S19 and T27 from ACE2 and S477 and F486 from RBD.

**Figure 3 ijms-23-13502-f003:**
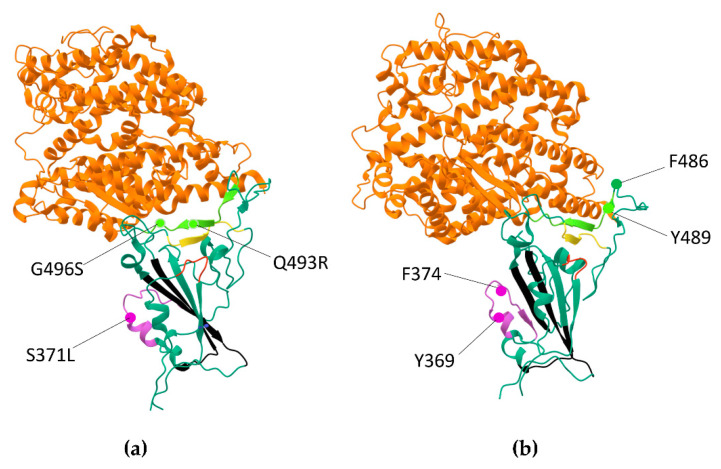
The 3D structure of Omicron BA.1 RBD bound to ACE2 according to 7T9L (https://www.rcsb.org/3d-view/7T9L/1, accessed on 20 June 2022). Omicron amyloidogenic tracts are as in Figure 2c. (**a**) Positions of Q493R, G496S, and S371L (R490, S493, and L368 in Omicron BA.1) are indicated as round shapes. Q493R and S371L are located in the longer amyloidogenic tracts and are critical for immune evasion. (**b**) Positions of Y369/F374 (Y366/F371 in Omicron) and F486/Y489 (F483/Y486 in Omicron) are indicated as round shapes. The 3D Omicron BA.1 RBD—ACE2 structure is rotated relative to (**a**).

**Table 1 ijms-23-13502-t001:** The RBDs and RBMs of different SARS-CoV-2 variants, their isoelectric points (pIs), charge at pH 7, and transmissibility.

SARS-CoV-2 Variant	RBD			RBM			Transmissibility (Average R0 Increase)
	RBD	pI	Charge at pH 7	RBM	pI	Charge at pH 7	
Wuhan-Hu-1 (WT)	323–531	8.27	3.9	438–506	7.94	0.8	1
Alpha	320–528	8.27	3.9	435–503	7.94	0.8	~29% increased over WT [69]
Beta	320–528	8.42	4.9	435–503	8.83	2.8	~25% increased over WT [69]
Epsilon	323–531	8.42	4.9	438–506	8.45	1.9	18.6–24% increased over WT [70]
Gamma	323–531	8.42	4.9	438–506	8.83	2.8	~38% increased over WT [69]
Delta	321–529	8.57	5.9	436–504	8.85	2.8	~97% increased over WT [69]
Omicron BA.1	320–528	8.70	7.2	435–503	9.55	6.1	~3.2 times more than Delta [71,72]
Omicron BA.2	320–528	8.70	7.2	435–503	9.55	6.1	~1.4 times more than BA.1 [72]

**Table 2 ijms-23-13502-t002:** Amyloidogenicity of Spike RBDs of various SARS-CoV-2 variants.

RBD	Amyloidogenicity * (%)
SARS_CoV (310–517)	26.9%
Delta (321–529)	29.7%
Epsilon (323–531)	29.7%
Wuhan-Hu-1 (323–531)	34.4%
Alpha (320–528)	34.4%
Beta (320–528)	34.4%
Gamma (323–521)	34.4%
Omicron BA.1 (320–528)	38.3%
Omicron BA.2 (320–528)	40.7%
HCoV-NL63 (481–616) **	47.1%

* The amyloidogenicity was calculated as described in the Section 4. ** The Spike RBD of HcoV-NL63 was determined as in [76].

## Data Availability

Not applicable.

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
