# Peer review of "The Increased Amyloidogenicity of Spike RBD and pH-Dependent Binding to ACE2 May Contribute to the Transmissibility and Pathogenic Properties of SARS-CoV-2 Omicron as Suggested by In Silico Study"

_ijms, 2022, doi:10.3390/ijms232113502_

Round 1
Reviewer 1 Report
Authors have performed in silico study of spike RBD of three variants of SARS-CoV-2 with ACE2 receptor at different pH and discussed the correlation of transmissibility and pathogenic properties of SARS-CoV-2 variants. They also discussed amyloidogenicity property of SARS-CoV-2 variants using in silico study. Although authors have presented and discussed the results, it requires more data to corroborate the conclusions.
1) Authors need to include other SARS-CoV-2 variants for MD simulations at different pH to understand role of pH in transmissibility of virus.
2) Authors need to provide more data from pH dependent MD simulations, like free energy change in steered MD at different pH between different variants, Residue wise interactions, RMSD etc. Based on current provided data, it is hard to conclude the role of pH.
Reviewer 2 Report
The authors describe two extended amyloidogenic sequences formed during SARS CoV2 mutations and the emergence of Omicron strains, and emphasize the importance of the functions of the mutated amino acids found in these sequences. For example, they analyzed one of the extended amyloidogenic sequences that spans the B2 sheet and surrounding loops and matches the key ACE2 binding site. In addition to the above, they show that amino acids mutated in SARS CoV2 Omicron strains include more basic amino acids and elaborate how this affects binding at different pH values. They structured their manuscript in the form of an article combining results and discussion.
I cannot recommend publishing the manuscript in this form for the following reasons:
1) In its current form, out of the 200 lines of the 2. Results and Discussion section (lines: 145-342), the author's results are presented with only 20 lines: 179-181; 195-199; 215-221; 245-246; 251-252; 273-275; 304-306. Most of these are still observations, not primary results. That does not seem like enough for a primary science article. In general, I think that this manuscript would work better in the form of a review. Within the review form, it is also possible to display part of your own results, which is even desirable.
2) An unacceptable series of errors in the preparation of work. Authors should check and correct all errors during the possible new preparation of the manuscript, and not only those that the reviewer specifically mentioned - of which there are already too many - because they refer literally to each of the few images of the results.
i. Line 176 ‘BFB‘ – typo
ii. Line 192 ‘RMB’ - typo
iii. Mark in Figure 1A only the amino acids that are explained in the text. For example, name only those 9 that are on the interaction surface of ACE2 and RBD, and show the rest with a black circle. Or highlight them with a different color. At least write, … while others, like T478K, are located on the periphery (line 157)’. The way the image is currently displayed is confusing.
iv. Please use clearer labels in Figure 1B (legend below the image). Be sure (MAJOR comment) to supplement the image with the necessary usual marks (such as stars) that would support statistical significance and lines (197-199) which reads: ‘However, the difference is statistically significant with Delta (at both speeds, 0.1 and 0.05 Å/ps) only in the protonated form and not at pH 7.’
v. Picture 2C - Error in color code - they show gray and write cyan. 'The secondary structure of Spike RBD is shown in cyan under the sequence.' Line 264. The text should be corrected and the colors for the backbone between 2B and 2C should be adjusted.
vi. Line 279: ‘Omicron amyloidogenic tracts are as in Figure 3C’. Figure 3C does not exist. They mean 2C.
3) The introduction should be more clearly structured, contain only basic information, but at the same time contain references to key manuscripts that precede the research, so that the scientific novelty of the work can be better assessed. There is almost no clear distinction between Section 1. Introduction and Section 2. Results and Discussion. E.g.:
i. Why there are no studies dealing with the influence of pH on RBD in the introduction in order to highlight the novelty. References like Cui et al comparing Omicron Spike in pH 5.5 and 7 conditions should not be found only in the results. The introduction must clearly indicate the current state of the art with regard to pH.
ii. Several times throughout the manuscript the authors repeat the idea that the pH in the upper and lower respiratory tract is different and that this coincides with better or worse binding of Omicron strains, highlighting differences in pI caused by mutations in Omicron. The authors also mention the proteases furin and TMPRSS2 in several places, and only note the existence of other proteases. Even when they claim that omicron is less dependent on TMPRSS2 they do not mention other proteases. It would be expected that they described cathepsins that have pH-dependent cleavage properties and that have been shown to affect SARS CoV 2 infection.
iii. The current structure of the five paragraphs in 1. Introduction is cca: 1. processing by proteases; 2. Amyloid deposition caused by SARS-CoV-2; 3. amyloid properties, now of protein S itself: 4 again about proteases 5. results and ideas. Why are the 1st and 4th paragraphs that are related to the same topic not together?
4) When revising the manuscript, it would be desirable to remove sentences that are duplicated (eg Line 204-206 ‘Omicron can easily spread in the upper respiratory tract, but damage less lungs and endothelium because the pH is less than optimal for its binding to ACE2 there.’, represents the third repetition of identical information within the same paragraph.); remove sentences that do not appear later anywhere in the results (eg line 253 Therefore, it is of great interest whether the increased amyloidogenicity of the binding interface may contribute to the kinetics of RBM binding to the receptor); remove sentences that do not directly address the manuscript (at the author's choice), because there is currently a lot of information from different scientific groups and some parts are difficult to follow.
Round 2
Reviewer 1 Report
Authors have managed to address first comment but not second comment which is very important.
2) Authors need to provide more data from pH dependent MD simulations, like free energy change in
steered MD at different pH between different variants, Residue wise interactions, RMSD etc. Based on
current provided data, it is hard to conclude the role of pH.
Authors reply - "Thank you. We have added such information. The initial coordinates of all systems were taken from the
Protein Data Bank (https://www.rcsb.org/). The 15 Å layer of water molecules was added near protein
structures. A collisional thermostat was used during all MD-experiments to temperature maintenance
near 300 K. The main integrity step was 0.001 ps-1. Main characteristics of 144 MD-experiments are
available at the site: http://oka.protres.ru/protres_sars. Video of pulling the RBD domain of the Wuhan
strain from ACE2 is available at the http://oka.protres.ru/protres_sars/Video/Expand-6m0j.avi, of
Omicron domain is available at the http://oka.protres.ru/protres_sars/Video/Expand-7t9l.avi. Also we
have added Table 1 with Potential energy between RBD and ACE2 domains after 4-ns relaxation."
I do not see any of the analysis included in the manuscript. These analysis are very important to make sure the conformational change is not due to artifacts of MD but it's real. Authors should show these analysis throughout time series like Figure 1C. Without those analysis it is hard to conclude the effect of pH. Also, authors should upload those mentioned videos in supporting information.
Reviewer 2 Report
I believe that the paper has been significantly improved and is now on the verge of publishing quality. I suggest the following corrections to the manuscript:
Problems in lines 139-142: a) up to that point in the text, the term RBM has not yet been defined; b) 'some disorder’ is not really scientific and it would be better to extract half a sentence of key information from colleagues who published the manuscript; finally c) sentence - I quote (electrostatic changes occurred in the Omicron RBD when compared to the prototype Wu…) is not at all clear. At different pH? to the same amino acids? Otherwise, it is obvious that there will be changes when it comes to a different sequence.
Figure 1 a is clearer. Why are some (eg S375F) yellow?
Line 217 (major), I Quote: The lower the potential energy value, the greater the force needed to tear one domain from another. In light of this, can you explain the data in the second column of Table 2 where the result for Omicron is between the Wuhan and Delta results? Shouldn't it show the strongest binding in the protonated form compared to the other strains?
Line 219??? Figure 1c. Not 3c.
246 – Please remove (that is characteristic of the upper respiratory tract). That same information is written in line 249. And then it is repeated again.
Lines 280-304 (major) It must be rewritten. To make it clear why something is in figure a, something in figure b. To make the point of what is being said clear. It is commendable that minor information has been removed, but this is the end of the results section of the manuscript and is almost impossible to read with understanding.
Line 420 - as part of the newly updated text, remind once again that it is an area with increased amyloidogenicity, because otherwise the continuation is not very clear.
Round 3
Reviewer 1 Report
Authors have addressed all the comments.